# Novel Mechanically Fully Decoupled Six-Axis Force-Moment Sensor

**DOI:** 10.3390/s20020395

**Published:** 2020-01-10

**Authors:** Chyi-Yeu Lin, Anton Royanto Ahmad, Getnet Ayele Kebede

**Affiliations:** 1Department of Mechanical Engineering, National Taiwan University of Science and Technology, Taipei 106, Taiwan; jerrylin@mail.ntust.edu.tw (C.-Y.L.); D10703814@mail.ntust.edu.tw (A.R.A.); 2Center for Cyber-Physical System, National Taiwan University of Science and Technology, Taipei 106, Taiwan; 3Taiwan Building Technology Center, National Taiwan University of Science and Technology, Taipei 106, Taiwan

**Keywords:** cross beam, cross-talk, mechanical decoupling, six-axis force/moment sensor, strain gauges

## Abstract

In this study, a novel six-axis force/moment (F/M) sensor was developed. The sensor has a novel ring structure comprising a cross-beam elastic body with sliding and rotating mechanisms to achieve complete decoupling. The unique sliding and rotating mechanisms can reduce cross-talk effects caused by minimized structural interconnection. The forces *F_x_*, *F_y_*, and *F_z_* and moments *M_x_*, *M_y_*, and *M_z_* can be measured for the six-axis F/M sensors according to the elastic deformation of strain gauges attached to the cross beam. Herein, we provide detailed descriptions of the mathematical models, model idealizations, model creation, and the mechanical decoupling principle. The paper also presents a theoretical analysis of the strain based on Timoshenko beam theory and the subsequent validation of the analysis results through a comparison of the results with those obtained from a numerical analysis conducted using finite element analysis simulations. The sensor was subjected to experimental testing to obtain the maximum cross-talk errors along the following six axes under different loadings (the errors are presented in parentheses): *F_x_* under *SM_y_* (2.12%), *F_y_* under *SM_x_* (1.88%), *F_z_* under *SM_z_* (2.02%), *M_x_* under *SF_z_* (1.15%), *M_y_* under *SF_x_* (1.80%), and *M_z_* under *SF_x_* (2.63%). The proposed sensor demonstrated a considerably improved cross-talk error performance compared with existing force sensors.

## 1. Introduction

Six-axis force/moment (F/M) sensors enable robots to obtain information about three-dimensional forces (*F_x_*, *F_y_*, and *F_z_*) and moments (*M_x_*, *M_y_*, and *M_z_*). According to the type of transducer used, six-axis F/M sensors can be classified into the following categories: strain gauge, piezoelectric, capacitor, and optical sensors. Most sophisticated six-axis F/M sensors use strain gauges. In a strain gauge sensor, the elastic body in the sensor transfers the applied load to the bonded strain gauge; the gauge then measures elastic deformation. Simultaneously achieving high sensitivity and decoupled outputs with a low cross-talk error in six-axis F/M sensors is challenging.

Decoupled six-axis F/M sensors proposed by previous studies can be classified into two groups: mechanically decoupled sensors and sensors using decoupling algorithms. In a parallel three-dimensional force sensor proposed in a previous study, mechanical decoupling was implemented to reduce the influence of dimensional coupling and improve sensing accuracy [1]. However, most existing six-axis F/M sensors lack complete integration. Therefore, instead of using mechanical decoupling methods, researchers have developed various decoupling algorithms using simple and convenient methods, such as linear regression [2] and the least squares method (LSM) [3,4,5], and advanced methods, including neural networks, the shape-form-motion approach [6], support vector regression (SVRs) [7,8], and least squares support vector machines (LS-SVMs) [9].

Regarding the development of the structures of F/M sensors, particularly the elastic body, design is critical to ensure optimal sensor performance. Many researchers have designed elastic body structures in the form of E-type membranes [10], T-shaped bars [11], Stewart platform [12], parallel plates [13], parallel beams [14], and Maltese crossbeams [15]. In a previous study, a cross-shaped double-hole structure was improved by considering structural errors resulting from the inaccuracy of the sensor body and by performing signal conditioning for noise reduction [16]. A previously proposed sensor involving a Maltese cross element with rigid inner and outer flanges can measure specific components of external forces; however, it has low sensitivity to horizontal forces (*F_x_* and *F_y_*), which subsequently causes considerable cross coupling [11]. 

A study proposed a numerical shape optimization design for a mechanically decoupled six-axis F/M sensor [17]. Moreover, scholars have proposed optimization methods for validating structural analysis models using conditional numbers, static and dynamic stiffness, and strain gauge sensitivity as sensor design parameters. A study optimized the design of a thin-type four-axis F/M sensor for a robot finger by applying structural optimization techniques in which strain gauges were positioned using a response surface method and a suitable function [18]. In another study, four identical T-shaped bars were subjected to finite element analysis (FEA) along with design optimization to maximize measurement sensitivity [11]. Furthermore, studies [19,20] have presented the Stewart platform that can achieve a comprehensive index optimization of the structural parameters of a force sensor by using indices atlases and a genetic algorithm. In another study, the performance and structural design of a prestressed six-axis F/M sensor with double layers were validated by considering the optimization objective to obtain optimal structural parameters [21]. To optimize the design of a multi-axis force sensor (MFS) integrated in a humanoid robot foot structure, a study analyzed the design criteria and strain gauge sensitivity as the objective function for MFS numerical optimization [22]. However, developing an F/M sensor with high performance requires appropriate design criteria for the sensor’s body structure and a numerical optimization process to evaluate the sensor’s sensing accuracy in terms of cross coupling and measurement error.

A precise sliding and rotating clearance between the symmetrical grooves and elastic body is necessary to decouple a structure. Otherwise, the contact force between the elastic body and the groove’s sidewall affects the structure. This study presents a novel mechanically decoupled six-axis F/M sensor with a sliding and rotating cross-elastic beam structure afforded by a unique strain gauge configuration [23]. In addition, this paper describes the theoretical calculation of the deformation, stress, and strain of the proposed six-axis F/M sensor through analytical modeling. The decoupling relationship between the sliding and rotating mechanisms is also analyzed herein through cross-talk error performance. On the basis of a mechanical and analytical model of the sensor and parameters affecting the strain values for the six-axis F/M sensor, stress and strain distributions are described herein [24]. 

The rest of this paper is organized as follows. Section 2 presents the decoupled structural design and strain gauge configuration of the proposed six-axis F/M sensor. Section 3 describes the design requirements and analysis of the sliding and rotating structure of the F/M sensor. Section 4 presents the results and discussion, including a comparison of numerical and analytical solutions for the optimized design structure of the F/M sensor, the results of friction analysis, and the results of experimental validation. Finally, Section 5 provides the conclusions.

## 2. Decoupled Structural Design and Strain Gauge Configuration

This section presents the decoupled structural design and strain gauge configuration of the proposed mechanically decoupled six-axis F/M sensor.

### 2.1. Decoupled Structural Design

The sensor structure is composed of aluminum 7075-T6 with a Young’s modulus of 71.7 GPa and tensile yield strength of 503 MPa. Figure 1 illustrates the design of the six-axis F/M sensor with a sliding structure. The six-axis F/M sensor comprises four main parts, namely elastic body, top flange, upper rim, and lower rim. The elastic body consists of four elastic beams with a connection box in the middle. The parameters *l*, *b*, and *h* represent the length, width, and height, respectively, of the four elastic beams, and *d* represents half of the connection box’s width or length. Table 1 presents the dimension parameters used in this paper. The top flange attaches the tool and transfers forces and moments to the elastic body. The upper and lower rims serve as the elastic body (sensor housing). The lower rim has two functions, serving as an end effector connector and as a base for the elastic body.

As presented in Figure 2, the sliding mechanism of the elastic body has two degrees of freedom on each side. Four ends of the elastic beam are cylindrical, enabling the sleeve bearings to hold onto the elastic body. These sleeve bearings enable translation and rotation along the axis. This sliding mechanism is selected because of its ability to generate low cross-talk errors [23].

### 2.2. Strain Gauge Arrangements

A double parallel strain gauge is attached to each of the surfaces of each elastic beam, as shown in Figure 3. Every four strain gauges are arranged in a Wheatstone full bridge circuit. Six Wheatstone bridge circuits were developed on the basis of Equation (1) presented in a previous study [23]. This equation was selected as a reference in the present study because of the use of double parallel strain gauges [23]. In contrast to most single- and serial-based studies [17,24,25,26], the present study used an strain gauge (DY43-3/350) from Hottinger Baldwin Messtechnick (HBM) as a transducer, with its resistance being 350 Ω.

The attachment positions of the strain gauges are determined on the basis of three considerations: the maximum strain, isotropy of the material involved, and prevention of non-linearity [15]. FEA can be implemented to analyze normal strains in the middle of the elastic beam. To achieve the results presented in Figure 4, the maximum load should be applied in gradual increments until the material’s yield stress is reached. The force can be maintained below the maximum yield stress to maintain the deformation in an elastic state, to avoid hysteresis, and to ensure a linear relationship between stress and strain. Because the structure is symmetrical, the force or moment applied in both the X and Y directions would be similar. Figure 4 shows that the strain is non-linear near both ends of the rectangular elastic beam. To achieve a high strain and to avoid non-linearity, all strain gauges should be separated by a distance of 3 mm.
(1)$${\mathrm{SG}}_{\mathrm{Fx}}=C_{1j}=\left(\frac{1}{4}\right)\left(\left({\mathrm{SG}}_{28}-{\mathrm{SG}}_{31}\right)+({\mathrm{SG}}_{15}-{\mathrm{SG}}_{12}\right))\;\;{\mathrm{SG}}_{\mathrm{Fy}}=C_{2j}=\left(\frac{1}{4}\right)\left(\left({\mathrm{SG}}_{8}-{\mathrm{SG}}_{3}\right)+({\mathrm{SG}}_{19}-{\mathrm{SG}}_{24}\right))\;{\mathrm{SG}}_{\mathrm{Fz}}=C_{3j}=\left(\frac{1}{4}\right)\left(\left({\mathrm{SG}}_{1}-{\mathrm{SG}}_{6}\right)+({\mathrm{SG}}_{18}-{\mathrm{SG}}_{21}\right))\;{\mathrm{SG}}_{\mathrm{Mx}}=C_{4j}=\left(\frac{1}{4}\right)\left(\left({\mathrm{SG}}_{25}-{\mathrm{SG}}_{30}\right)+({\mathrm{SG}}_{13}-{\mathrm{SG}}_{10}\right))\;\;{\mathrm{SG}}_{\mathrm{My}}=C_{5j}=\left(\frac{1}{4}\right)\left(\left({\mathrm{SG}}_{17}-{\mathrm{SG}}_{22}\right)+({\mathrm{SG}}_{5}-{\mathrm{SG}}_{2}\right))\;{\mathrm{SG}}_{\mathrm{Mz}}=C_{6j}=\left(\frac{1}{4}\right)\left(\left({\mathrm{SG}}_{32}-{\mathrm{SG}}_{27}\right)+({\mathrm{SG}}_{16}-{\mathrm{SG}}_{11}\right))$$

## 3. Structural Analytic Solution

Plastic deformation occurs when the F/M-induced stress on a flexure exceeds the yield strength of the flexure. The structural part of the elastic body is crucial for the F/M sensor design. In the F/M sensor structure design proposed in this paper, the strains of the elastic beam structure are fully analyzed using Timoshenko beam theory [27]. In this design, symmetry with respect to loads in the *x* and *z* directions is determined through a bending analysis for a single cantilever module. This section describes the design requirements and the analysis of the sliding and rotating structure of the F/M sensor.

In Timoshenko beam theory, illustrated by Figure 5, the bending moment is denoted by *M* and the shearing force is denoted by *Q* [27]. Let *ϕ* be the angle caused by bending and *γ* be the angle caused by shear. Beam deformation characteristics are described by two unknown parameters: the translational displacement (*ω*) and angular displacement (*φ*) of any cross section in terms of *x*. The coordinate value of any point on the beam can be expressed as follows:(2)$$M\left(x\right)=-EI\frac{d\varphi \left(x\right)}{dx}\;Q\left(x\right)=kGA\left(\frac{d\omega \left(x\right)}{dx}-\varphi \left(x\right)\right)$$

The F/M equilibrium equations for the infinitesimal element of the beam are based on Equation (2). The following ordinary differential equations can be obtained after linking Equation (2) with the infinitesimal element of the beam:(3)$$\frac{d}{dx}\left[kGA\left(\frac{d\omega \left(x\right)}{dx}-\varphi \left(x\right)\right)\right]=0\;\frac{d}{dx}\left[kGA\left(\frac{d\omega \left(x\right)}{dx}-\varphi \left(x\right)\right)-\left(EI\frac{d\varphi \left(x\right)}{dx}\right)\right]=0$$
where *EI* is flexural rigidity, *κ* is a constant depending on the shape of the cross section of the beam, A is the cross-sectional area, and *G* is the modulus of rigidity. Therefore, $I=\frac{bh^{3}}{12},\;$and A=bh is the cross-sectional area of the elastic beam.

The two basic formulas in Timoshenko beam theory are presented in Equations (2) and (3) and can be used to calculate an analytical solution of *ω*(*x*) and *φ*(*x*) if sufficient boundary conditions are provided. The strain value (*ε*) of any point on a beam can be calculated using Equation (4) after the analytical solution of the angular displacement *φ*(*x*) is obtained:(4)$$\varepsilon \left(x,z\right)=-z\frac{d\varphi \left(x\right)}{dx}$$
where *z* is any point along a beam’s *z* coordinate; it also indicates the distance between the point and the neural plane.

The following sections present the proposed mechanical model for elastic beams when pure forces or pure moments are applied to the elastic body. The sections also present the effects of these forces or moments on the strain distributions of the elastic body.

### 3.1. Under Applied Force F_x_ or F_y_

When a pure force *F_x_* is applied to the F/M sensor (Figure 6), the axial forces are transmitted through elastic beams *DH* and *BF* to produce a sliding motion in the elastic body of the non-contact force sensor (i.e., no bending deformation occurs). Concurrently, bending deformation occurs in beams *AE* and *CG*. *A*′, *B*′, and *C*′ represent the displaced positions of *A*, *B*, and *C*, respectively. Δ*x*, $\Delta A{A^{\prime }}_{Fx}$, or *ω* represents the displacement of the rectangular box (i.e., the displacement node of *A* in elastic beam *AE*). The bending of beam *AE* is the same as that of beam *CG*. The bending stress levels observed on the strain gauges of the two horizontal beams are equal. We can assume that the output of the strain gauges (SG_15_, SG_28_ and SG_12_, SG_31_) regarding the bending stress levels of the two elastic beams (*AE* and *CG*) are equal.

The force equilibrium equation of the rectangular box can be derived as expressed in Equation (5), where$\;F_{AFx}$ is the shear force on cross-elastic beam *AE.*
(5)$$F_{x}=2F_{AFx}$$

$\Delta A{A^{\prime }}_{Fx\;}$ can be obtained using a combination of Equation (2) and boundary conditions of Equation (3) observed from beam *AE* in Figure 6 and expressed as follows:(6)$$\omega \;\left(x\right)=\Delta A{A^{\prime }}_{Fx}=\frac{l^{3}F_{AFx}}{3EI}+\frac{F_{AFx}l}{kGA}$$

By substituting Equation (5) into Equation (6) and then substituting the result into Equation (3) and then Equation (4), we obtain$\;\varepsilon_{F_{x}}\;$as a function of *F_x_* as follows:(7)$$\varepsilon_{F_{x}}\left(x,z\right)=\frac{zF_{x}x}{2EI}$$

### 3.2. Under Applied Force F_z_

When a pure force $F_{z}$ is applied to the F/M sensor (Figure 7), bending deformation occurs on all elastic beams (*AE*, *BF*, *CG*, and *DH*). The outputs of the strain gauges (SG_1_, SG_18_ and SG_6_, SG_21_) regarding the bending stress levels for the four elastic beams are equal. The geometrical characteristics of the deformation of the beams are illustrated in the free body diagram in Figure 7. All elastic beams have fixed boundary conditions at nodes *E*, *F*, *G* and *H*, corresponding to simply supported beams. Δ*z*, $F_{AFz},\;$or *ω* represents the displacement of the rectangular box (i.e., the displacement node of *A* in elastic beam *AE*).

The force equilibrium equations of the elastic body are expressed as follows, where $\;F_{AFz}$ represents the shear force in elastic beam *AE*.
(8)$$F_{z}=4F_{AFz}$$

As displayed in Figure 7, $\Delta A{A^{\prime }}_{Fz}\;$represents the vertical distance between node $A^{\prime }$ and node E. $\Delta A{A^{\prime }}_{Fz}\;$can be derived using Equation (2) and the boundary conditions in Equation (3) obtained from elastic beam *AE* in Figure 7, and it can be expressed as follows: (9)$$\omega \;\left(x\right)=\Delta A{A^{\prime }}_{Fz}=\left(\frac{l^{3}F_{AFz}}{3EI}+\frac{F_{AFz}l}{kGA}\right)$$
where$\;\varepsilon_{F_{z}}\;$represents the strain at any point on the elastic beam under *F_z_* loading, which is derived by substituting Equation (8) into Equation (9), then into Equation (3), and finally into Equation (4):(10)$$\varepsilon_{F_{z}}\left(x,z\right)=\frac{zF_{z}x}{4EI}$$

### 3.3. Under Applied Moment M_x_ or M_y_

When a pure moment $M_{x}$ is applied to the F/M sensor, elastic beams *DH* and *BF* produce a pure rotating motion following the elastic body of the non-contact force sensor (i.e., no bending deformation is produced). Concurrently, bending deformation occurs in elastic beams *AE* and *CG*. The bending of elastic beam *AE* is similar to that of elastic beam *CG*. Figure 8 presents the deformation of the elastic body under the applied moment $M_{x}$, where $\Delta \theta_{Mx}$ or *φ* represents the angle of rotation of the rectangular box with respect to x-axis in global coordinates. The outputs of the strain gauges (SG_13_, SG_25_, SG_10_, and SG_30_) regarding the bending stress levels of the two elastic beams (*AE* and *CG*) are equal.

The F/M equilibrium equation of the elastic body is expressed as follows:(11)$$M_{x}=M_{AMx}-F_{AMx}d$$
where $M_{AMx}\;$is the bending moment on the upper surface of beam *AE,* and $F_{AMx}=\frac{M_{AMx}}{l}.$

The geometric characteristics of the deformed elastic body under the applied *M_x_* can be calculated using Equation (2) and the boundary conditions in Equation (3) obtained from the deformed characteristics of beam *AE*:(12)$$\varphi \;\left(x\right)=\Delta \theta_{Mx}=\frac{-kGAl^{3}F_{AMx}-3EIlF_{AMx}}{3kGAEI\left(l+d\right)}$$
where *d* represents the distance from the center to node *A* on the rectangular box.

The strain value under *M_x_* loading at any point on the elastic beam, represented by$\;\varepsilon_{M_{x}}$, can be derived by substituting Equation (11) into Equation (12) and then into Equation (3), followed by substituting the results into Equation (4):(13)$$\varepsilon_{M_{x}}\left(x,z\right)=\frac{zM_{x}x}{2EIl}$$

### 3.4. Under Applied Moment M_z_

When a pure moment $M_{z}$ is applied to the F/M sensor, bending deformation occurs on all elastic beams (*AE*, *BF*, *CG,* and *DH*). The outputs of the strain gauges (SG_16_, SG_32_ and SG_11_, SG_27_) regarding the bending stress levels corresponding to the four elastic beams are equal. The geometric characteristics of deformation are illustrated in the free body diagram in Figure 9. All elastic beams have fixed boundary conditions at nodes *E*, *F*, *G,* and *H*, corresponding to simply supported beams. $\Delta \theta_{Mz}\;$or *φ* represents the angle of rotation of the rectangular box with respect to the z-axis in global coordinates under the applied moment$\;M_{z}$.

The F/M equilibrium equation of the square convex under $M_{z}\;$can be expressed as follows:(14)$$M_{z}=-4M_{AMz}-F_{AMz}d$$
where$\;F_{AMz}$ and $M_{AMz\;}\;$are the shear force and bending moment of the upper surface of *AE*, respectively, and$\;M_{AMz}=\;F_{AMz\;}l$.

The angle of rotation $\Delta \theta_{Mz}\;$derived using Equation (2) and the boundary conditions in Equation (3) obtained from beam *AE* can be expressed as follows:(15)$$\varphi \;\left(x\right)=\Delta \theta_{Mz}=\frac{kGAl^{3}M_{z}-3EIlM_{z}}{12kGAEI{\left(l+d\right)}^{2}}$$

Similarly, substituting Equation (14) into Equation (15), then into Equation (3), and finally into Equation (4) can yield the strain value $\varepsilon_{M_{z}}$ under *M_z_* loading:(16)$$\varepsilon_{M_{z}}\left(x,z\right)=\frac{zM_{z}x}{4EIl}$$

## 4. Results and Discussion

During the design of a six-axis F/M sensor, several design factors should be considered, including the elastic body’s specific force geometry, sensing element, material, sensing bridge design, stress-strain limits, measuring force limits, and size limitations. The strain can be determined using Timoshenko beam theory and validated using FEA simulation. 

### 4.1. Numerical Simulations

As presented in Figure 10, this study simulated an FEA model to analyze the design parameters for the sensor structure in order to confirm the strains calculated using the derived equations under *F_x_* and *F_z_* and *M_x_* and *M_z_*. The ANSYS program was used for the numerical solutions. The strain gauges for the *F_x_*, *F_y_*, *F_z_*, *M_x_*, *M_y_*, and *M_z_* components were attached 3 mm from the edge of the rectangular box along the direction of the elastic beam’s length.

### 4.2. Comparison of Numerical Solution with Analytical Solution

In the analytical experiment, the Timoshenko equation was modified because it was designed for fixed or pin-type supports only. Because the edges were designed to slide at the end of the elastic beams in this study, constants must be added to the Timoshenko equation. These constants were added according to Figure 4. These constants were derived at points where the strain was equal to zero and started to turn negative. Consequently, the Timoshenko equation was adjusted and is expressed as follows:(17)$$\varepsilon_{F_{x}}\left(x,z\right)=-z\frac{F_{x}}{2EI}\left(x-9.75\right)\;\varepsilon_{F_{z}}\left(x,z\right)=-z\frac{F_{z}}{4EI}\left(x-9.75\right)\;\varepsilon_{M_{x}}\left(x,z\right)=-z\frac{M_{x}}{2EIl}\left(x-12.5\right)\;\varepsilon_{M_{z}}\left(x,z\right)=-z\frac{M_{z}}{4EIl}\left(x-12.5\right)$$

The analysis was conducted using the desired forces: *F_x_*/*F_y_* = 400 N, *F_z_* = 800 N, *M_x_*/*M_y_* = 30 Nm, and *M_z_* = 30 Nm. The simulation was performed using ANSYS, and normal strain data were obtained at the center of the surface of the elastic beams, as illustrated in Figure 11. An analytical solution was also obtained using the modified Timoshenko equation and is presented alongside the results of the numerical analysis in Figure 11. These results demonstrate that the results of the modified Timoshenko equation are similar to the numerical results. However, the final tail of the graph for the numerical results is shown to indicate non-linear strains, whereas the graph for the Timoshenko equation results shows only a straight line. These results agree with those of studies that have compared Timoshenko results with numerical results [22,27].

This study focused on the strain gauge placement position on rectangular elastic beams at a distance of 3 mm from the center box. Table 2 presents a comparison of the numerical simulation results and the analytical solutions. When *M_y_* = 30 Nm, the largest error was 2.51%. The corresponding error in [25] was 7.7%, which implies that our modified Timoshenko equation is superior.

This structure is then tested to quantify the cross-talk errors generated by Equation (18) as shown in Table 3. This equation is well known among researchers, to name a few studies undertaken by [14,17,28] which using this equation is used to make cross-talk error. The largest cross-talk error generated in this study was 0.35%. In an ANSYS-based study conducted previously that involved the same configuration as that used in the present study [23], a cross-talk error of 11% was obtained. Therefore, our proposed sensor with a sliding structure is superior to sensors without sliding structures. Another study [29] proposed a sensor with sliding structures and double-type elastic cross beams; for this sensor, the maximum error generated was 0.89%. However, the difference between this sensor and that proposed in the present study is the strain gauge configuration. The strain gauge configuration in the sensor proposed in the present study is considerably more suitable for sliding mechanisms.
(18)$$S_{ij}=\left(\frac{C_{ij}}{C_{ii}}\right)\times 100\%$$

*S_ij_* is the measured strain, with *i* = 1, …, 6 representing the measurement provided by the strain gauge bridge and *j* = 1, …, 6 representing the specified load on the force sensor. For example, *S*_11_ represents the reading provided by strain gauge bridge *F_x_* (SG FX) when *F_x_* is applied to the force sensor (Equation (1)). When *i* ≠ *j*, a cross-talk error may occur on the force sensor. For example, *S*_12_ indicates the reading provided by SG *F_x_* when *F_y_* is applied.

### 4.3. Optimization

The Timoshenko equation could result in relatively low errors and could facilitate the execution of optimization processes. Sensor optimization is typically performed to obtain a more acceptable range of measurements and accurate and durable sensor products. The objective of optimization is to attain specific strain levels on each axis to maximize sensitivity. In this study, optimization was performed using sequential quadratic programming, a widely employed method [24,30,31]. To protect the strain gauge under an elastic state, the specific strains for each axis were set as follows: *S_FX_* = *S_FY_* = *S_FZ_* = 200, *S_MX_* = *S_MY_* = 600, and *S_MZ_* = 300. The loading levels applied in each direction were as follows: *F_x_* = *F_y_* = 400 N, *F_z_* = 800 N, *M_x_* = *M_y_* = *M_z_* = 30 Nm. The design variable [*b*,*h*] was determined such that γ (attain factor) was minimized; this could be achieved using the following equation:(19)$$\frac{b}{2}\times \frac{F_{x}}{2EI}\left(6.75\right)-200\gamma \leq 200\;\frac{h}{2}\times \frac{F_{z}}{4EI}\left(6.75\right)-200\gamma \leq 200\;\frac{h}{2}\times \frac{M_{x}}{2EIl}\left(9.5\right)-600\gamma \leq 600\;\frac{b}{2}\times \frac{M_{z}}{4EIl}\left(9.5\right)-300\gamma \leq 300\;8.0\leq b\leq 12.0\;mm\;8.0\leq h\leq 12.0\;mm$$

The design optimization process resulted in the elastic beam width (*b*) and thickness (*h*) listed in Table 4.

Because this study yielded new optimized structural parameters, additional simulations were required. One of the simulations was conducted using the modified Timoshenko equation, and the other was conducted through a numerical analysis. As illustrated in Figure 12, the results obtained using the modified Timoshenko equation were similar to the numerical analysis results. Moreover, the results for *F_x_*/*F_y_* and *F_z_* were similar; therefore, the plot for *F_z_* cannot be observed in the figure. According to the optimized structural dimensions, strain analysis was performed on the specified strain gauge configuration throigh both numerical simulation and numerical analysis. As presented in Table 5, the errors generated for *F_x_*/*F_y_* and *F_z_* were nearly negligible, but those for *M_x_*/*M_y_* and *M_z_* were substantial. However, compared with the 7.7% deviation reported by a previous study [26], the result of the present study is superior.

A cross-talk error analysis was performed for the sensor with the optimized structure. As revealed in Table 6, the largest error was observed for *F_y_* (0.51%), which is relatively high compared with the results of related studies. However, the error is controllable and this is more favorable than that reported by a previous study [29] that utilized a sliding framework and achieved a maximum error of 0.89%.

### 4.4. Friction

The simulation and analysis results presented in the preceding sections are suitable for structures in which antifriction bearings are used between the elastic body and the upper and lower rims (housing part). When a dry-sleeve bearing is used, which is small, low maintenance, and without a lubricant, it may cause minimum friction, with the corresponding friction coefficients being approximately 0.01–0.20. Therefore, some inaccuracies may be produced. Table 7 shows the results of a friction analysis conducted on a sleeve bearing with a friction coefficient of 0.20. The design of this analysis was based on the initial design presented in Table 1. Values presented in boldface type in Table 7 represent errors pertaining to the non-frictional strain analysis. The highest error achieved was 0.58% under *F_z_*, and the friction increased the cross-talk error. The highest cross-talk error observed for the *M_x_* sensor under *F_x_* was −0.53% which is still less than the previous study [23].

### 4.5. Experimental Validation

Experimental testing is necessary to obtain a clear evaluation of a novel model. Accordingly, this study conducted an experiment for the proposed sensor. A prototype of the sliding force sensor (Figure 13) was manufactured on the basis of the initial design presented in Table 1. A dry-sleeve bearing composed of bronze material was used in the force sensor. The first step in the experiment was to calibrate the sensor (Figure 14) using the LSM. After the calibration, the sensor was tested to determine its performance. Table 8 presents the average output error and cross-talk error. The values in boldface type in the table represent the error readings for the applied forces under a specific load. Maximum measurement error was 2.00% and the maximum cross-talk error occurred when *M_z_* loading on *SF_x_* was 2.63%.

The measurement error obtained in this paper showed some error difference compared to the previous study [23], which could have been 1.78%, even with the same strain gauge arrangement. But if we equate the cross-talk error, the present study shows a significant improvement from 4.78% to 2.63%. Nonetheless, if we equate it with studies [28] that use LSM to get calibration matrix, the measurement error they obtain is 6.85% and the crosstalk error is 4.85%. Therefore, the development of the force sensor compared to the manufactured prototypes has been achieved.

Looking at the simulation test, this model is expected to achieve the best performance so far. The real challenge is to make it so that the result would be the same with the simulation. The manufacturing process of the sensor has always faced a structural error of the body and the bonding process effect of the strain gauge leads to a minor error from analysis to experimental testing.

## 5. Conclusions

This study proposes a novel decoupled F/M sensor with a unique strain gauge configuration based on boundary conditions on the beam configuration. An FEA model was used to determine design criteria for achieving minimum stiffness with a high force sensitivity and low cross-sensitivity. The elastic body of the sensor is symmetrical and constrained by upper and lower beam covers accommodating sliding and rotating mechanisms. The cross-sensitivity of this F/M sensor is considerably reduced by appropriate selection of suitable structural design parameters.

The findings of this research are outlined as follows:The elastic beams of the force sensor address sliding and rotating mechanisms through a new strain gauge configuration, which leads to a considerably superior cross-talk error performance compared with other force sensors. The maximum cross-talk error for *F_x_* occurred when *SM_y_* was 0.41%, *F_y_* occurred when *SM_x_* was −0.42%, *F_z_* occurred when *SM_x_* was −0.51%, *M_x_* occurred when *SF_y_* was −0.39%, *M_y_* occurred when *SF_x_* was 0.25%, and *M_z_* occurred when *SF_y_* was −0.17%.By using the modified Timoshenko formula, we ensured that the dimensions of the force sensor matched the set target.Experimental validation indicated favorable results, with the maximum cross-talk error being <3% [14]. The maximum cross-talk error for *F_x_* occurred when *SM_y_* was 2.12%, *F_y_* occurred when *SM_x_* was 1.88%, *F_z_* occurred when *SM_z_* was 2.02%, *M_x_* occurred was *SF_z_* was 1.15%, *M_y_* occurred when *SF_x_* was 1.80%, and *M_z_* occurred when *SF_x_* was 2.63%. However, we achieved a minimum accuracy of 98.00%.It can be said that the strain gauge configuration in study [23] is very suitable for sliding mechanisms.

Based on these results, we might suggest for further study the use of a better manufacturing process and some advanced decoupling algorithm.

## Figures and Tables

**Figure 1 sensors-20-00395-f001:**
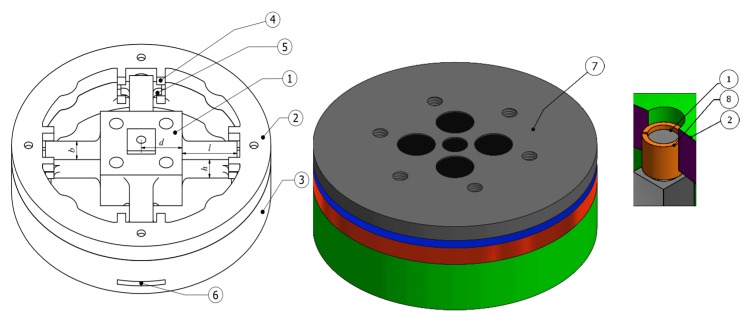
Six-axis force/moment (F/M) sensor design and structural components: (**1**) elastic body, (**2**) upper rim (sensor housing), (**3**) lower rim (sensor housing), (**4**) rim stopper, (**5**) beam stopper, (**6**) cable hole, (**7**) top flange, and (**8**) sleeve bearing.

**Figure 2 sensors-20-00395-f002:**
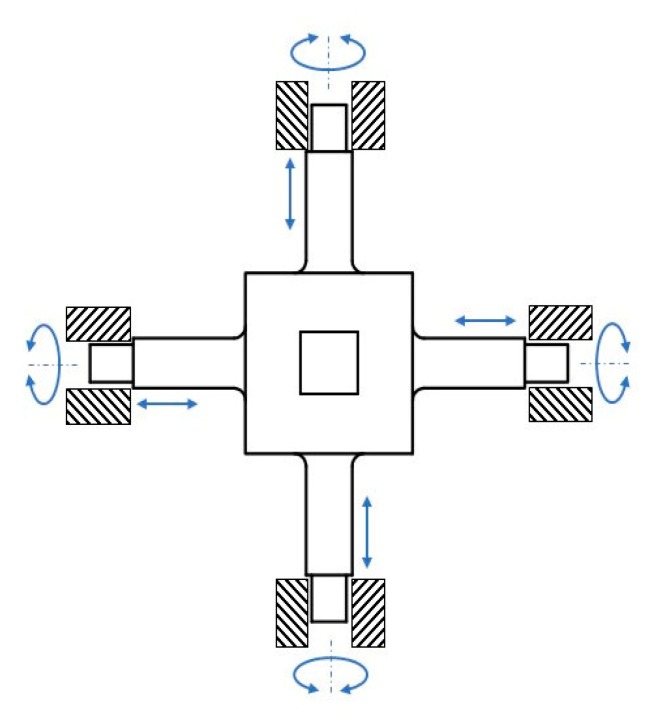
Sliding mechanism of the elastic body.

**Figure 3 sensors-20-00395-f003:**
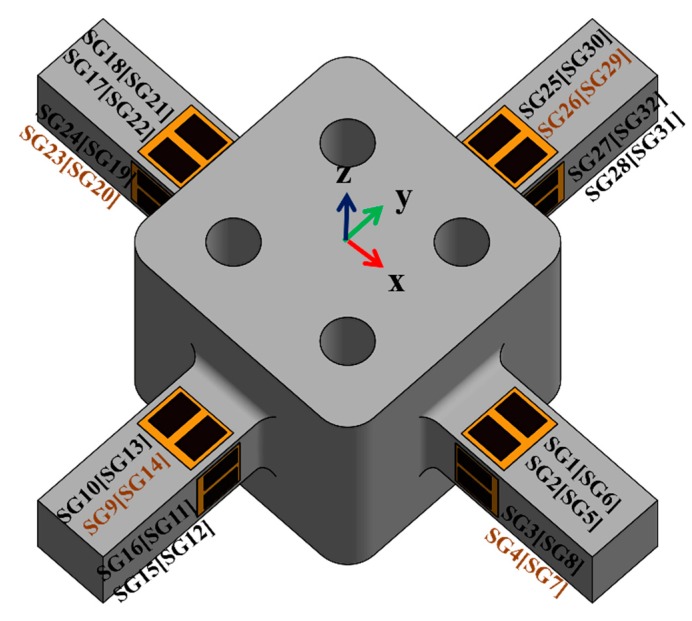
Configuration of strain gauges [23].

**Figure 4 sensors-20-00395-f004:**
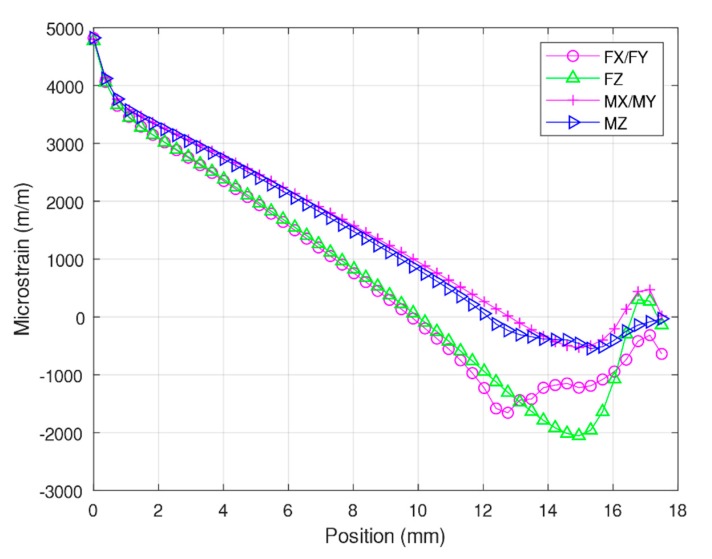
Normal strain on the middle of elastic beam.

**Figure 5 sensors-20-00395-f005:**
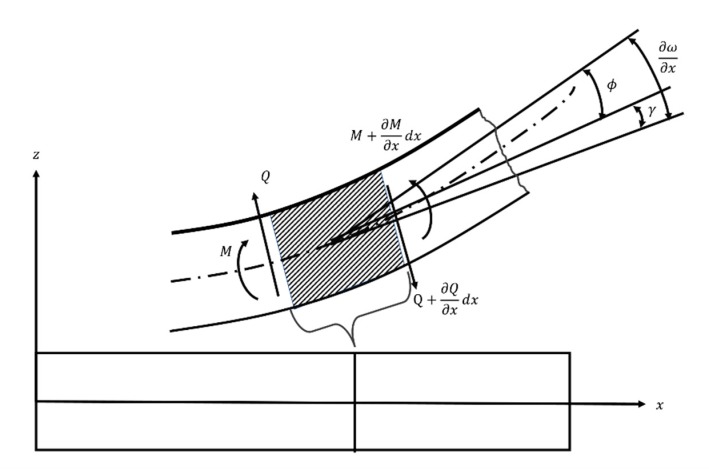
A Timoshenko beam and element of the beam.

**Figure 6 sensors-20-00395-f006:**
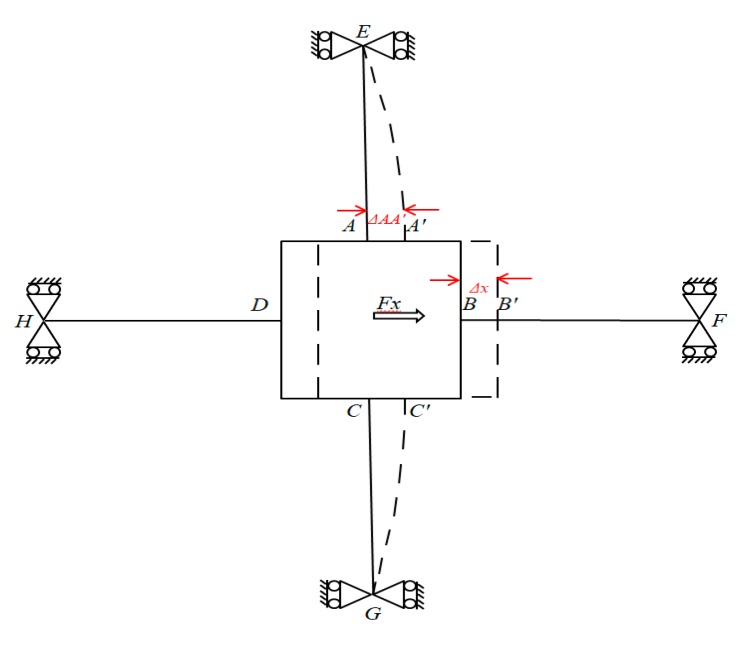
Deformation of the elastic body under force *F_x_.*

**Figure 7 sensors-20-00395-f007:**
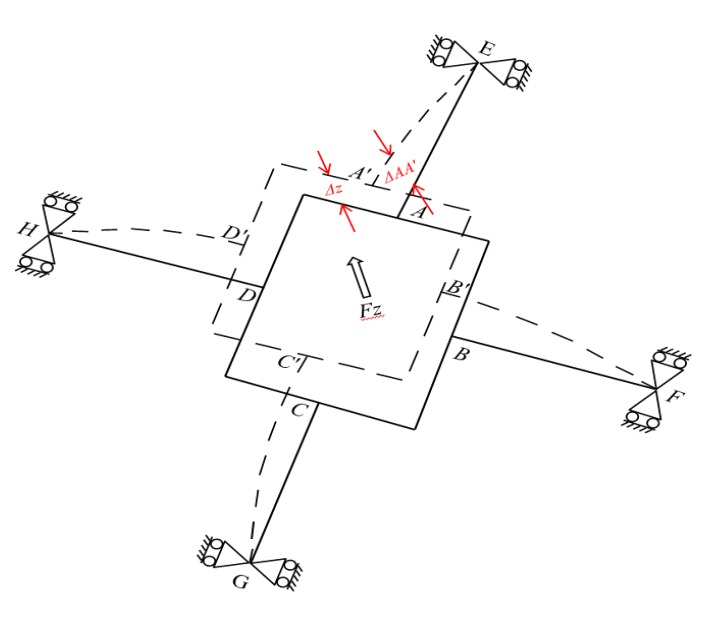
Deformation of the elastic body under force $F_{z}$

**Figure 8 sensors-20-00395-f008:**
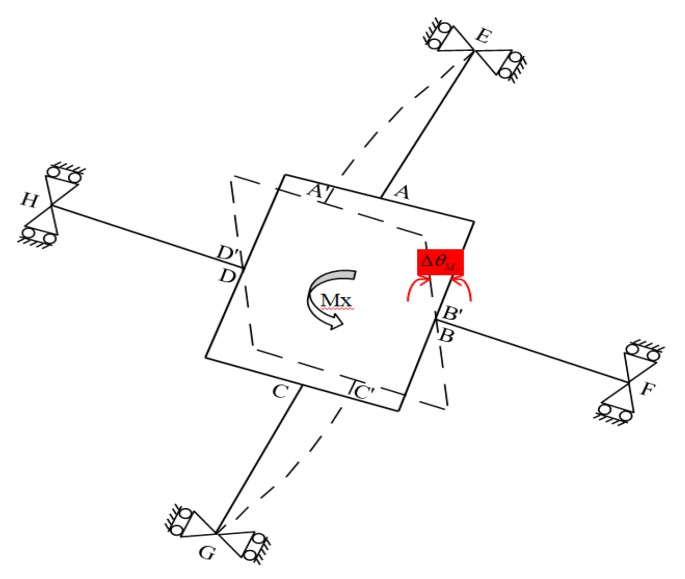
Deformation of the elastic body under moment $M_{x}$

**Figure 9 sensors-20-00395-f009:**
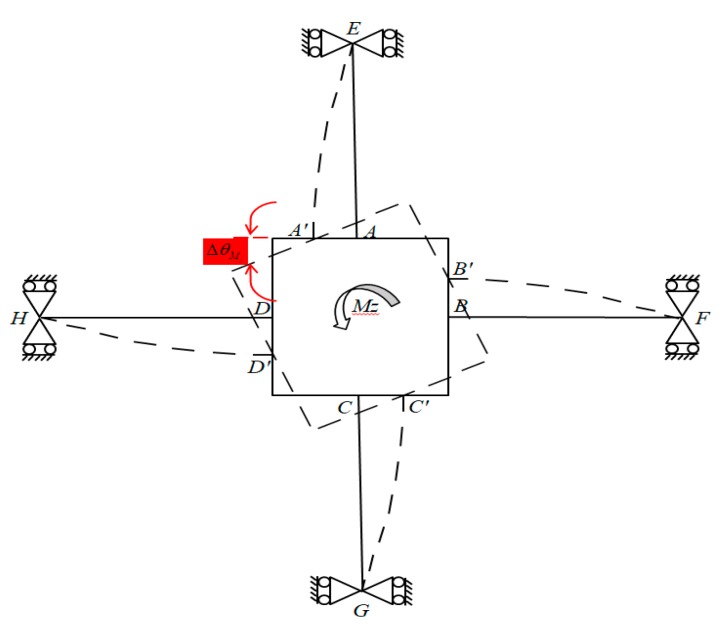
Deformation of the elastic body under moment$\;M_{z}$

**Figure 10 sensors-20-00395-f010:**
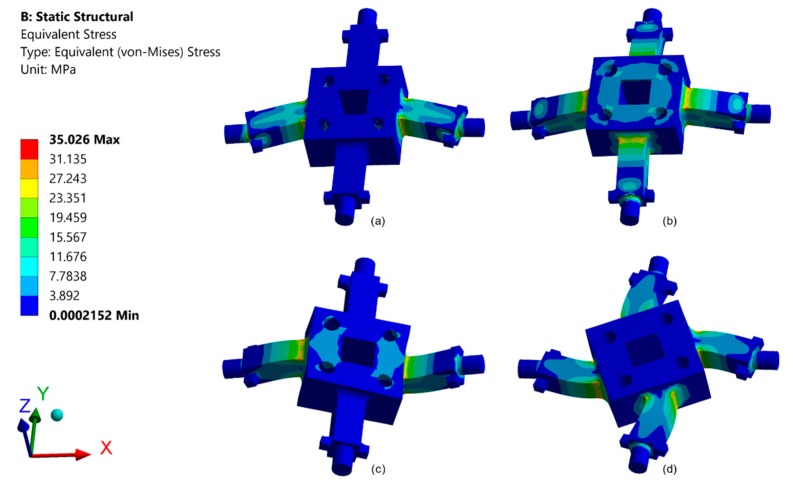
Strain analysis of the elastic body under the applied F/M: (**a**) pure load *F_x_*/*F_y_*, (**b**) pure load *F_z_*, (**c**) pure load *M_x_*/*M_y_*, (**d**) pure load *M_z_*.

**Figure 11 sensors-20-00395-f011:**
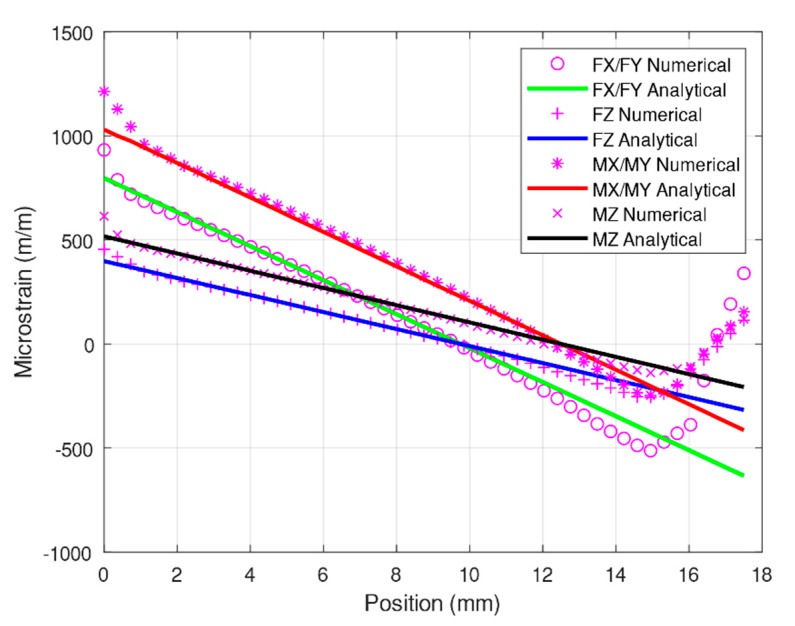
Comparison of numerical and analytical solutions under *F_x_*/*F_y_, F_z_*, *M_x_*/*M_y_*, and *M_z_* of elastic beams.

**Figure 12 sensors-20-00395-f012:**
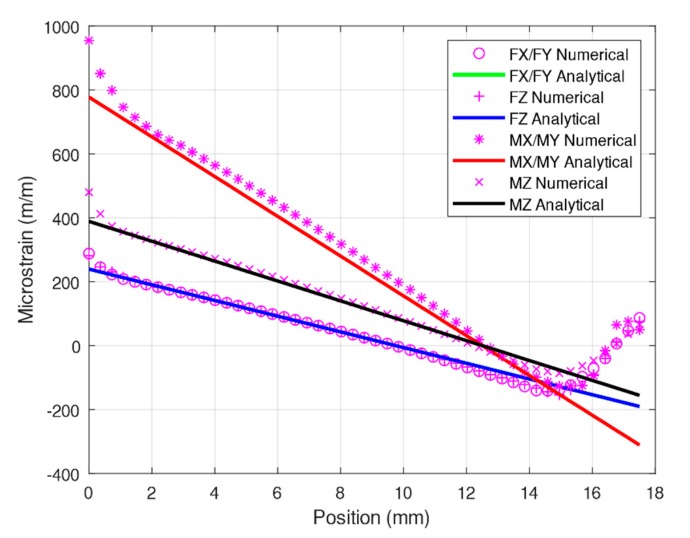
Comparison of numerical and analytical solutions under *F_x_/F_y_, F_z_, M_x_/M_y_,* and *M_z_* of the optimized elastic beams.

**Figure 13 sensors-20-00395-f013:**
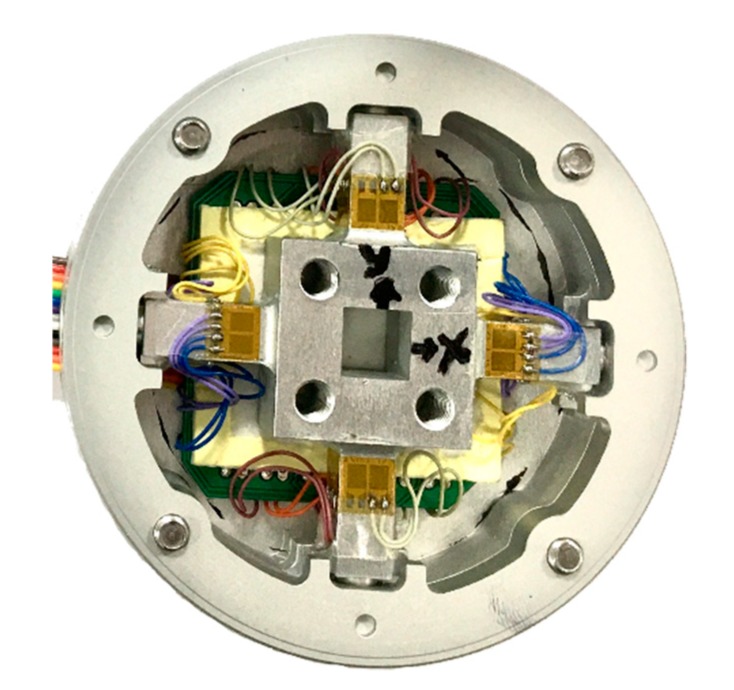
Proposed sliding F/M sensor.

**Figure 14 sensors-20-00395-f014:**
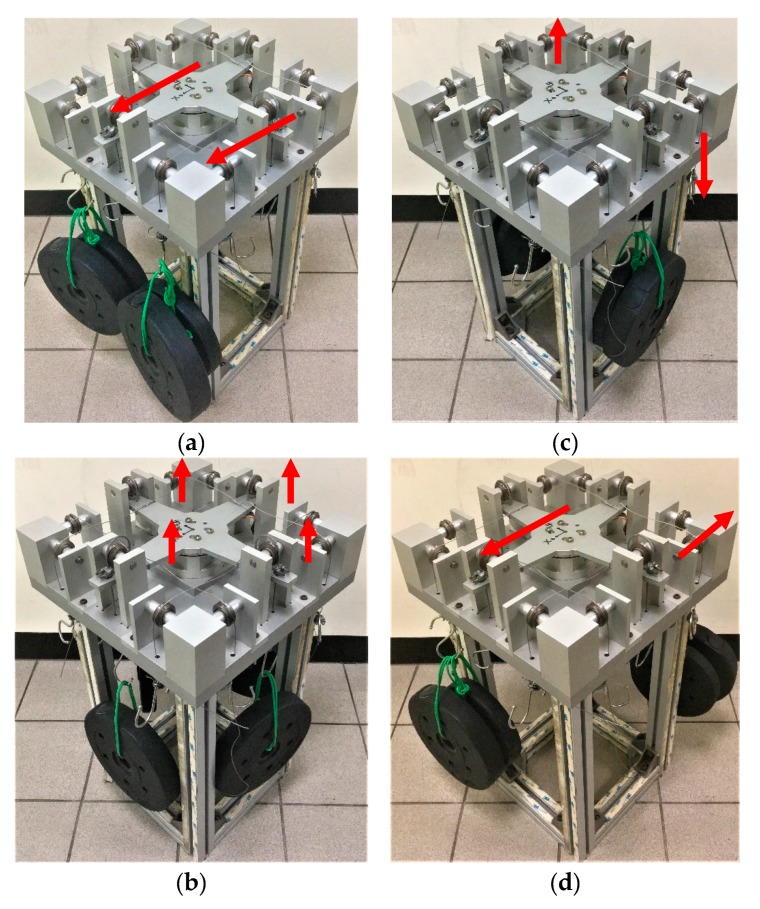
Schematic of the applied loading conditions on the jig: (**a**) *F_x_* or *F_y_*, (**b**) *F_z_*, (**c**) *M_x_* or *M_y_*, and (**d**) *M_z_* [23].

**Table 1 sensors-20-00395-t001:** Dimension parameters of the sensor.

Parameters	Dimension (mm)
Length of elastic beam	*l* = 17.5
Height of elastic beam	*h* = 8.0
Width of elastic beam	*b* = 8.0
Half of width of rectangular box	*d* = 16

**Table 2 sensors-20-00395-t002:** Comparison of strains on elastic beam between numerical simulation and analytical solution.

	Numerical Simulation	Analytical Solution	Errors
*F_x_*	218	221	1.35%
*F_y_*	218	221	1.26%
*F_z_*	219	221	0.87%
*M_x_*	806	787	−2.37%
*M_y_*	807	787	−2.51%
*M_z_*	394	393	−0.24%

**Table 3 sensors-20-00395-t003:** Cross-talk readings of the finite element analysis (FEA) results.

	LOAD *F_x_* 400 N	LOAD *F_y_* 400 N	LOAD *F_z_* 800 N	LOAD *M_x_* 30 Nm	LOAD *M_y_* 30 Nm	LOAD *M_z_* 30 Nm
*SF_x_*	-	0.01%	−0.19%	0.04%	0.14%	0.09%
*SF_y_*	0.00%	-	0.03%	−0.35%	−0.03%	0.07%
*SF_z_*	−0.01%	0.35%	-	−0.12%	−0.14%	0.15%
*SM_x_*	−0.28%	−0.31%	−0.17%	-	−0.01%	−0.06%
*SM_y_*	0.33%	−0.26%	−0.24%	−0.01%	-	−0.19%
*SM_z_*	0.05%	0.07%	0.09%	0.09%	0.01%	-

**Table 4 sensors-20-00395-t004:** Optimal dimensions of the sensor.

Parameters	Dimension (mm)
Length of elastic beam	*l* = 17.5
Height of elastic beam	*h* = 8.8
Width of elastic beam	*b* = 8.8
Half of width of rectangular box	*d* = 16

**Table 5 sensors-20-00395-t005:** Comparison of strain on elastic beam between numerical simulation and analytical solution in the optimized structure.

	Numerical Simulation	Analytical Solution	Errors
*F_x_*	165	166	0.55%
*F_y_*	166	166	−0.33%
*F_z_*	167	166	−0.79%
*M_x_*	623	591	−5.18%
*M_y_*	625	591	−5.47%
*M_z_*	303	295	−2.48%

**Table 6 sensors-20-00395-t006:** Cross-talk readings of the FEA results of the optimized sensor.

	LOAD *F_x_* 400 N	LOAD *F_y_* 400 N	LOAD *F_z_* 800 N	LOAD *M_x_* 30 Nm	LOAD *M_y_* 30 Nm	LOAD *M_z_* 30 Nm
*SF_x_*	-	−0.01%	0.05%	0.13%	0.25%	−0.17%
*SF_y_*	0.00%	-	0.11%	−0.39%	0.02%	−0.07%
*SF_z_*	0.07%	−0.12%	-	−0.07%	−0.02%	0.13%
*SM_x_*	0.00%	−0.42%	−0.51%	-	−0.01%	−0.12%
*SM_y_*	0.41%	−0.08%	−0.09%	−0.01%	-	0.03%
*SM_z_*	−0.25%	−0.08%	−0.08%	0.07%	−0.01%	-

**Table 7 sensors-20-00395-t007:** Cross-talk and error readings of the FEA results of the sensor with friction.

	LOAD *F_x_* 400 N	LOAD *F_y_* 400 N	LOAD *F_z_* 800 N	LOAD *M_x_* 30 Nm	LOAD *M_y_* 30 Nm	LOAD *M_z_* 30 Nm
*SF_x_*	**0.58%**	0.00%	0.19%	−0.09%	0.15%	−0.27%
*SF_y_*	0.01%	**0.53%**	0.01%	−0.33%	0.08%	−0.07%
*SF_z_*	0.05%	0.40%	**0.57%**	−0.02%	0.11%	0.22%
*SM_x_*	−0.53%	−0.32%	0.11%	**−0.10%**	0.01%	−0.09%
*SM_y_*	0.31%	−0.43%	−0.01%	0.00%	**0.27%**	−0.30%
*SM_z_*	−0.33%	−0.05%	−0.03%	0.00%	−0.03%	**0.16%**

**Table 8 sensors-20-00395-t008:** Absolute average error analysis.

	LOAD *F_x_*	LOAD *F_y_*	LOAD *F_z_*	LOAD *M_x_*	LOAD *M_y_*	LOAD *M_z_*
*SF_x_*	**0.69%**	1.81%	2.02%	1.02%	1.80%	2.63%
*SF_y_*	1.34%	**1.13%**	1.40%	1.12%	0.72%	1.80%
*SF_z_*	1.59%	1.64%	**0.95%**	1.15%	0.69%	0.94%
*SM_x_*	1.66%	1.88%	1.37%	**2.00%**	0.65%	0.90%
*SM_y_*	2.12%	1.41%	1.85%	0.46%	**1.50%**	1.52%
*SM_z_*	1.94%	1.39%	1.59%	0.81%	0.95%	**0.94%**

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
