# Peer review of "Novel Mechanically Fully Decoupled Six-Axis Force-Moment Sensor"

_sensors, 2020, doi:10.3390/s20020395_

Round 1

Reviewer 1 Report

In this manuscript, the author analyzed the strains of the elastic beam structure by Timoshenko beam theory and obtained the analytical solution. Subsequently, the author validated the aforementioned results by FEA simulation. However, some results are not clearly presented, which must be improved.

In introduction, some important references about the decouple of multi-dimensional force sensors are not cited and evaluated as follows:[1] A novel self-decoupled four degree-of-freedom wrist force/torque sensor. Measurement, 2007, 40(9-10): 883-891.  [2] Fast estimation of strains for cross-beams six-axis force/torque sensors by mechanical modeling. Sensors, 2013, 13(5): 6669-668. [3] Decoupling Strategy of Multi-dimensional Force Sensor Based on LS-SVM and αth-order Inverse System Method[C]//2007 8th IEEE International Conference on Electronic Measurement and Instruments., 2007: 4-378-4-381. [4] Shape optimization of a mechanically decoupled six-axis force/torque sensor[J]. Sensors and Actuators A: Physical, 2014, 209: 41-51. [5] A force sensor with five degrees of freedom using optical intensity modulation for usage in a magnetic resonance field[J]. Measurement Science and Technology, 2013, 24(4): 045101.

Vector diagrams should be as substitutes for fuzzy pictures (Figure 1, 3, 4, 5, 10, 11, 12) in this manuscript. A partial enlarged detail of the sliding and rotating structure should be appended to Figure 1. How to attain equation (17)? Why do you need to add constants to Timoshenko formula? Page 11, line 263: Units should be added to the desired force.  Results in Table 2 and Figure 11 are not consistent, and this seems to be wrong. Similarly, there are also the same problems in Table 5 and Figure 12. How to obtain the crosstalk results in Table 3 from the finite element analysis? Please give adequately description.

Reviewer 2 Report

The paper presents a design for a 6 axis force-moment sensor. The basic elastic element is a cross-beam structure which is well known (see e.g. ref. [12]). The exact novelty of the presented design should be explained more clearly in the introduction section. It seems that the novelty consists of 2 aspects:

1) The cross-beam structure is allowed to slide and rotate at the ends of the beams. Other researchers have avoided sliding structures as the measurement will become dependent on friction forces and usually other elastic suspensions are used.

2) Placement of the strain gauges is somewhat novel, but was presented by the authors in an earlier paper, ref [19].

The explanation and analysis of the design in sections 2, 3 and 4 is clear, however the paper lacks an analysis of the influence of friction forces. As the sliding mechanism seems to be the main novelty this should be discussed.

The paper would be much stronger if measurement results on a manufactured structure were included so that a quantitative comparison with other designs and evaluation of friction force influence can be made.

Some minor remarks:

Line 17 in the abstract seems to be a repetition of line 14.

Round 2

Reviewer 1 Report

This manuscript developed a six dof force/torque sensor with a novel ring structure. The design of the sensor is an interesting. The author has revised the manuscript according to my last comments, and the quality of the paper is improved lot. I think it can be accepted for publication.